# High-Frequency guided Curriculum Learning for Class-specific Object Boundary Detection

## Abstract

This work addresses class-specific object boundary extraction, i.e., retrieving boundary pixels that belong to a class of objects in the given image. Although recent *ConvNet*-based approaches demonstrate impressive results, we notice that they produce several false-alarms and misdetections when used in real-world applications. We hypothesize that although boundary detection is simple at some pixels that are rooted in identifiable high-frequency locations, other pixels pose a higher level of difficulties, for instance, region pixels with an appearance similar to the boundaries; or boundary pixels with insignificant edge strengths. Therefore, the training process needs to account for different levels of learning complexity in different regions to overcome false alarms. In this work, we devise a curriculum-learning-based training process for object boundary detection. This multi-stage training process first trains the network at simpler pixels (with sufficient edge strengths) and then at harder pixels in the later stages of the curriculum. We also propose a novel system for object boundary detection that relies on a fully convolutional neural network (FCN) and wavelet decomposition of image frequencies. This system uses high-frequency bands from the wavelet pyramid and augments them to *conv* features from different layers of FCN. Our ablation studies with *contourMNIST* dataset, a simulated digit contours from MNIST, demonstrate that this explicit high-frequency augmentation helps the model to converge faster. Our model trained by the proposed curriculum scheme outperforms a state-of-the-art object boundary detection method by a significant margin on a challenging aerial image dataset.

## 1 Introduction

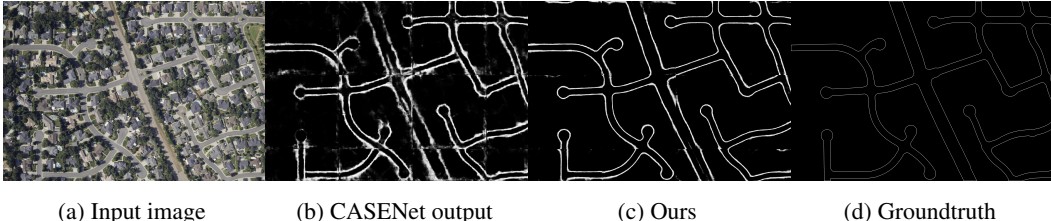

| (a) Input image | (b) CASENet output | (c) Ours | (d) Groundtruth |

Figure 1: Class-specific object boundary extraction: This work proposes a novel *ConvNet* architecture & a two-stage training scheme for class-specific object boundary estimation. This is a sample result from the model when applied for the problem of extracting road contours from aerial image tiles.

Class-specific object boundary extraction from images is a fundamental problem in Computer Vision (CV). It has been used as a basic module for several applications including object localization [Yu et al. (2018a); Wang et al. (2015)], 3D reconstruction [Lee et al. (2009); Malik & Maydan (1989); Zhu et al. (2018)], image generation [Isola et al. (2017); Wang et al. (2018)], multi-modal image alignment Kuse & Shen (2016), and organ feature extraction from medical images [Maninis et al. (2016)]. Inspired from the sweeping success of deep neural networks in several CV fields, recent works [Yu et al. (2017); Acuna et al. (2019); Yu et al. (2018b)] designed *ConvNet*-based architectures

for object boundary detection and demonstrated impressive results. However, we notice that the results from these methods, as shown in Figure 1b, still suffer from significant false-alarms and misdetections even in regions without any clutter. We hypothesize that although boundary detection is simple at some pixels that are rooted in identifiable high-frequency locations, other pixels pose a higher level of difficulties, for instance, region pixels with an appearance similar to boundary-pixels; or boundary pixels with insignificant edge strengths (ex: camouflaged regions). Therefore, the training process needs to account for different levels of learning complexity around different pixels to achieve better performance levels.

In classical CV literature, the different levels of pixel complexities are naturally addressed by decomposing the task into a set of sequential sub-tasks of increasing complexities [Leu & Chen (1988); Lamdan & Wolfson (1988); Kriegman & Ponce (1990); Ramesh (1995)]. Most often, boundary detection problem has been decomposed into three sub-tasks: (a) low-level edge detection such as Canny [Canny (1986)]; (b) semantic tagging/labeling of edge pixels [Prasad et al. (2006)] and (c) edge linking/refining [Jevtić et al. (2009)] in ambiguous regions. These approaches first solve the problem for simpler pixels (with sufficient edge strength) and then reason about harder pixels in the regions with ambiguous or missing evidence. However, with the advent of *ConvNets*, this classical perspective towards boundary extraction problem has been overlooked. New end-to-end trainable *ConvNets* have pushed the boundaries of state-of-the-art significantly compared to classical methods. However, we believe that classical multi-stage problem-solving schemes can help to improve the performance of *ConvNet* models. A parallel machine learning field, *Curriculum Learning* [Bengio et al. (2009)] also advocates this kind of multi-stage training schemes which train the network with a smoother objective first and later with the target task objective. These schemes are proven to improve the generalization of the models and convergences of training processes in several applications. Motivated by these factors, this work devises a *curriculum-learning* inspired two-stage training scheme for object boundary extraction that trains the networks for simpler tasks first (sub-tasks a and b) and then, in the second stage, trains to solve the more complex sub-task (c). Our experimental results on a simulated dataset and a real-world aerial image dataset demonstrate that this systematic training indeed results in better performances.

As mentioned already, the task of predicting object boundaries is mostly rooted in identifiable higher-frequency image locations. We believe that explicit augmentation of high-frequency contents to ConvNet will improve the convergence of training processes. Hence, this work designs a simple fully convolutional network (FCN) that takes in also high-frequency bands of the image along with the RGB input. Here in this work, we choose to use high-frequency coefficients from wavelet decomposition [Stephane (1999)] of the input image and augment them to *conv* features at different levels. These coefficients encode local features which are vital in representing sharp boundaries. Our empirical results convey that this explicit high-frequency augmentation helps the model to converge faster, especially in the first stage of curriculum learning.

In summary, our contributions in this work are the following:

- A novel two-stage training scheme (inspired from curriculum-learning) to learn class-specific object boundaries.
- A novel ConvNet augmented by high-frequency wavelets.
- A thorough ablation study on a simulated MNIST digit-contour dataset.
- Experiments with a challenging aerial image dataset for road contour extraction
- A real-world application of road contour extraction for aligning geo-parcels to aerial imagery.

**Related Work**: The problem of extracting object boundaries has been extensively studied in both classical and modern literature of CV. Most of the classical methods start with low-level edge detectors and use local/global features to attach semantics to the detected pixels. They, later, use object-level understanding or mid-level Gestalt cues to reason about missing edge links and occluded boundaries. The work by Prasad et al. (2006) used local texture patterns and a linear SVM classifier to classify edge pixels given by Canny edge detector. The work by Mairal et al. (2008) reasoned on low-level edges, but learned dictionaries on multiscale RGB patches with sparse coding and used the reconstruction error curves as features for a linear logistic classifier. The work of Hariharan et al. (2011) proposed a detector that combines low-level edge detections and semantic outputs from pre-trained object detectors to localize class-specific contours.

Several recent works Yang et al. (2016); Yu et al. (2017) adopted fully convolutional networks (FCN) for the task of semantic boundary extraction. The work by Yang et al. (2016) proposed a FCN-based encoder-decoder architecture for object contour detection. The work of Bertasius et al. (2015) first used VGG-based network to locate binary semantic edges and then used deep semantic segmentation networks to obtain category labels. More recently, the work of Yu et al. (2017) proposed an FCN architecture, CASENet, with a novel shared concatenation scheme that fuses low-level features with higher *conv* layer features. The shared concatenation replicates lower layer features to separately concatenate each channel of the class activation map in the final layer. Then, a $K$-grouped $1 \times 1$ *conv* is performed on fused features to generate a semantic boundary map with $K$ channels, in which the $k$-th channel represents the edge map for the $k$-th category. A few recent works Acuna et al. (2019); Yu et al. (2018b) integrated an alignment module into the network to account for noise in the contour labels. Most of these networks are trained in an end-to-end manner, using cross-entropy-based objective functions. These objectives treat all boundary pixels equally irrespective of the complexity around them. Unlike existing methods, we use explicit high-frequency augmentation to ConvNet and train it in a curriculum learning scheme that accounts for different levels of pixel complexities with two stages.

## 2 The proposed Curriculum Learning scheme

**Curriculum Learning & Multi-Stage Training**: Curriculum learning (CL) or multi-stage training schemes are motivated from the observation that humans and animals seem to learn better when trained with a curriculum like strategy: start with easier tasks and gradually increase the difficulty level of the tasks. A pioneering work of Bengio et al. (2009) introduced CL concepts to machine learning fields. This work proposed a set of CL schemes for the applications of shape recognition and language modeling; demonstrated better performance and faster convergence. This work established curriculum learning as a continuation method. Continuation methods [Allgower & Georg (2012)] start with a smoothed objective function and gradually move to less smoothed functions. In other terms, these methods consider a class of objective functions that can be expressed as,

$$C_\lambda(\theta) = (1 - \lambda)C_o(\theta) + \lambda C_t(\theta) \tag{1}$$

where $C_o(\theta)$ is smoother or simple objective and $C_t(\theta)$ is the target objective we wish to optimize. There are several ways to choose $C_o$; it can either be the same loss function as $C_t$ but solving the task on simpler examples, or be a proxy task simpler than the target task. In general, $\lambda$ is a (binary) variable that takes values zero or one. It is set to zero initially and later increases to one. The epoch where it changes from zero to one is referred to as the *switch epoch*.

**Curriculum Learning in CV**: CL-inspired training schemes are recently gaining attention in CV fields. Recently some popular CV architectures leveraged CL based schemes to improve model generalization and training stabilities. In FlowNet 2.0 [Ilg et al. (2017)] for optical flow prediction, simpler training data are fed into the network first and then the more difficult dataset. The object detection framework of Zhang et al. (2016) first trains simpler networks (proposal, refiner-nets) and then trains the final output-net in the end. Here we propose a two-stage CL scheme for object boundary detection methods.

**The proposed CL based training scheme for learning Object Boundaries**: ConvNets for class-specific object boundary detection are in general trained with multi-label cross entropy-based objectives [Yu et al. (2017)].

$$C_t(\theta) = - \sum_k \sum_p \left( \beta Y_k(p) \log \hat{Y}_k(p; \theta) + (1 - \beta)(1 - Y_k(p)) \log(1 - \hat{Y}_k(p; \theta)) \right) \tag{2}$$

where $\theta$ denotes the weights of the network; and $p$ and $k$ represent indices of pixel and class labels respectively. $\hat{Y}$ and $Y$ represent prediction and groundtruth label maps. $\beta$ is the percentage of non-edge pixels in the image to account for skewness of sample numbers [Yu et al. (2017)]. This objective treats all the contour pixels equally and does not account for the complexity of the task around them. Here, we consider this as the target objective function, $C_t$, that we wish to optimize.

We start the training, however, with a simpler task $C_o$. We believe the pixels with strong edge strength are easy to localize and semantically identify. Hence, we propose to solve the task around

those pixels in the first stage. We take element-wise multiplication between canny edge map of the input and dilated groundtruth label to prepare supervisory signal $Z$ for this stage.

$$Z = E_I \odot Y_D \tag{3}$$

where $E_I$ is the Canny edge map of image $I$ and $Y_D$ the dilated groundtruth map. This is as shown in Figure 2e.

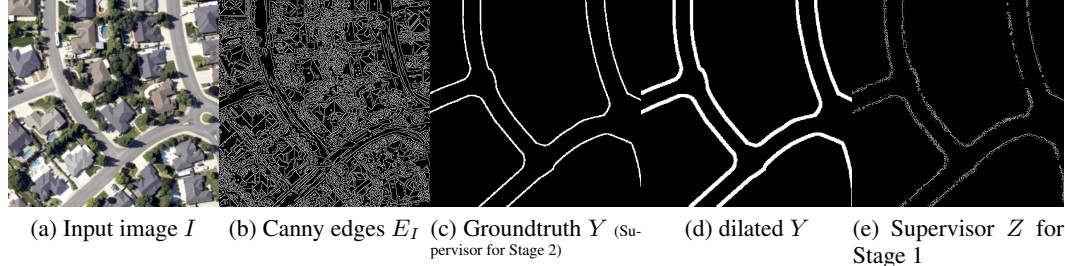

(a) Input image $I$    (b) Canny edges $E_I$    (c) Groundtruth $Y$ (Supervisor for Stage 2)    (d) dilated $Y$    (e) Supervisor $Z$ for Stage 1

Figure 2: Supervisory signal preparation for Stage 1

The objective function for this stage becomes;

$$C_o(\theta) = -\sum_k \sum_p \left( \beta Z_k(p) \log \hat{Z}_k(p; \theta) + (1 - \beta)(1 - Z_k(p)) \log(1 - \hat{Z}_k(p; \theta)) \right) \tag{4}$$

Since we use a dilated version of GT in preparing $Z$, it also contains some non-object contour pixels. However, these might be refined in the second stage of CL, when trained with $Y$ (Eq 2). Hence, the CL objective function in Eq 1 uses Eq 4 and Eq 2 as initial and target objective functions respectively. In the CL training scheme, we set the switch epoch as $T/2$, where $T$ is the total number of training epochs.

## 3 HIGH FREQUENCY AUGMENTED CONVNET

The first stage in the proposed CL-based training scheme is about learning to locate a class of high-frequency pixel locations and recognizing their semantic entity. We believe that explicit augmentation of high-frequency contents of the input image to the network will help learn faster. Hence, we design a novel architecture with explicit high-frequency augmentation that can make effective use of our multi-stage training scheme.

**Frequency Decomposition via Wavelet Transform (WT)**: Here we briefly introduce the main concepts of wavelet decomposition of the images [Stephane (1999)]. The multi-resolution wavelet transform provides localized spatial frequency analysis of images. This transform decomposes the given image into different frequency ranges, thus, permits the isolation of the frequency components introduced by boundaries into certain subbands, mainly in high-frequency subbands. The 2D wavelet decomposition of an image ($I \in R^{2M \times 2N}$) is performed by applying 1D low-pass ($\phi$) and high-pass ($\psi$) filters. This operation results in four decomposed subband images at each level, referred to as low-low ($W^{ll}$), low-high ($W^{lh}$), high-low ($W^{hl}$), and high-high ($W^{hh}$) wavelet coefficients. For instance, the single-level WT is defined as follows,

$$
\begin{aligned}
W_1^{ll}(i, j) &= \sum_k \sum_l I(2i + k, 2j + l)\phi(k)\phi(l) \\
W_1^{lh}(i, j) &= \sum_k \sum_l I(2i + k, 2j + l)\phi(k)\psi(l) \\
W_1^{hl}(i, j) &= \sum_k \sum_l I(2i + k, 2j + l)\psi(k)\phi(l) \\
W_1^{hh}(i, j) &= \sum_k \sum_l I(2i + k, 2j + l)\psi(k)\psi(l)
\end{aligned}
\tag{5}
$$

All the convolutions above are performed with stride 2, yielding a down-sampling of factor 2 along each spatial dimension. The WT results in four bands $\{W^{ll}, W^{lh}, W^{hl}, W^{hh}\} \in R^{M \times N}$ at first

level. A multi-scale wavelet decomposition successively performs Eq 5 on low-low frequency coefficients $\{.\}^{ll}$ from fine to coarse resolution. In this sense of resolution, decomposition in multi-resolution wavelet analysis is in analogy to the down-sampling steps in ResNet blocks [He et al. (2016)]. Moreover, it is worth noting that, while the low-frequency coefficients $\{.\}^{ll}$ store local averages of the input data, its high-frequency counterparts, namely $\{.\}^{lh}$, $\{.\}^{hl}$ and $\{.\}^{hh}$ encode local textures which are vital in recovering sharp boundaries. This motivates us to make use of the high-frequency wavelet coefficients to improve the quality of pixel-level boundary extraction. Throughout this paper, we extensively use the Haar wavelet for its simplicity and effectiveness to boost the performances of the underlying ConvNet. In this scenario, the Haar filters used for decomposition in Eq 5, are given by $\phi = (0.5, 0.5)$ and $\psi = (0.5, -0.5)$.

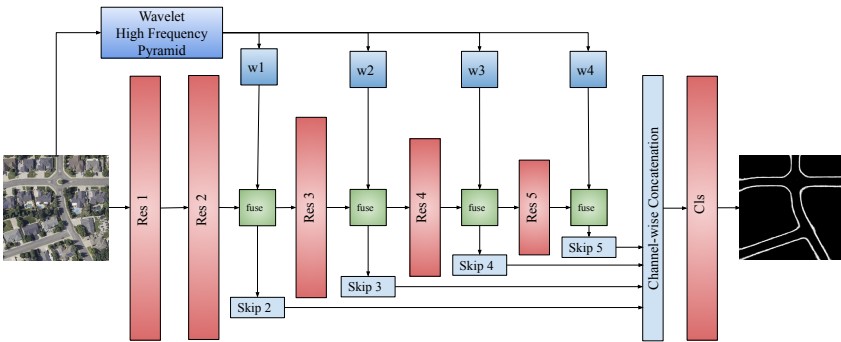

Figure 3: High-frequency augmented contour detection architecture.

**The proposed ConvNet**: Our architecture is as shown in Figure 3. We now explain the components of the network.

**ResNet-101 Backbone**: Several recent ConvNet architectures including CASENet use ResNet-101 as a basic building block. We use similar architecture with simple concatenation as our backbone architecture. ResNet-101 with five res blocks that uses 4 down-sampling steps (max-pool or stride 2). This results in 4 tensors of *conv* features at four scales: $\{F_i\}_{i=1,..,4}$

**Wavelets augmentation**: Multi-scale wavelet decomposition of the input image with four levels produce a pyramid of frequency bands. The spatial resolutions of these bands are analogous to *conv* features at each level. We select high frequency bands at each level of wavelet pyramid and concatenate in channel dimension of the *conv* features at corresponding level in the ResNet: $\{F_i \oplus W_i^{lh} \oplus W_i^{hl} \oplus W_i^{hh}\}_{i=1,...,4}$.

**Fuse blocks**: These *Fuse* module are designed to learn proper fusion schemes to fuse wavelets to *conv* features. These are based on $1 \times 1$ *conv* blocks that takes in $\{F_i \oplus W_i^{lh} \oplus W_i^{hl} \oplus W_i^{hh}\}$ and produce features to match next block's input dimensions.

**Skip modules**: Similar to *CASENet*, we design a set of skip connections that pass lower layer features to the classification layer. These skip modules uses up-sampling followed by $1 \times 1$ *conv* layers (with 64 filters). After up-sampling, these features from different resolutions are brought back to original input resolution. All these features are concatenated in the channel axis.

***cls* module**: This is a pixel-level classification module implemented with *conv* layer with $k$ filters, followed by a sigmoid layer to produce class-specific object boundary maps from concatenated features from different levels of ResNet. For the experiments provided in Section 4, we use $k = 1$ as we work with the problems of single class object contour extraction.

## 4 EXPERIMENTS

This section provides an extensive evaluation of our boundary extraction model and training scheme with a simulated dataset as well as a challenging real-world dataset. Our simulated dataset is prepared for a thorough ablation study without tolerating long training periods. Our real-world dataset is collected to train the proposed model for road contour extraction from aerial images, that is an essential component for the system we build in Section 5.

## 4.1 ABLATION STUDIES WITH MNIST DIGIT-CONTOURS

### 4.1.1 EXPERIMENTAL SETUP

**Dataset preparation**: The MNIST dataset [Deng (2012)] is popularly known toy dataset in machine learning, originally designed for image classification algorithms. Lately, it has been adapted to work with several tasks such as object detection [multiMNIST Eslami et al. (2016)], color image generation [colorMNIST Metz et al. (2016)], and Spatio-temporal representation learning [movingMNIST Srivastava et al. (2015)]. These adopted datasets have been used to understand the behavior of the models and training processes without tolerating long training periods. Similarly, we also simulate *contourMNIST* dataset using MNIST digits as objects-of-interest. The MNIST database contains 70000 gray-scale images of handwritten digits of resolution $28 \times 28$. The test images (10,000 digits) were kept separate from the training images (60,000 digits). For our study, we simulate $128 \times 128$-sized images with random digits placed on a random background sampled from Pascal-VOC dataset [Everingham et al. (2015)] (as shown in Figure 4). Corresponding pixel-level groundtruth labels for digits contours are also generated for each simulated image. To mimic human labeling noise in the groundtruth of real-world training data, we transform these GT labels (see Figure 4d) with randomly generated *tps* (thin-plate-splines) transformations. In this process, we prepare a train set of 5000 images and a validation set of 500 images. Training on this dataset for ten epochs takes just around 17 minutes for all the experiments in this section.

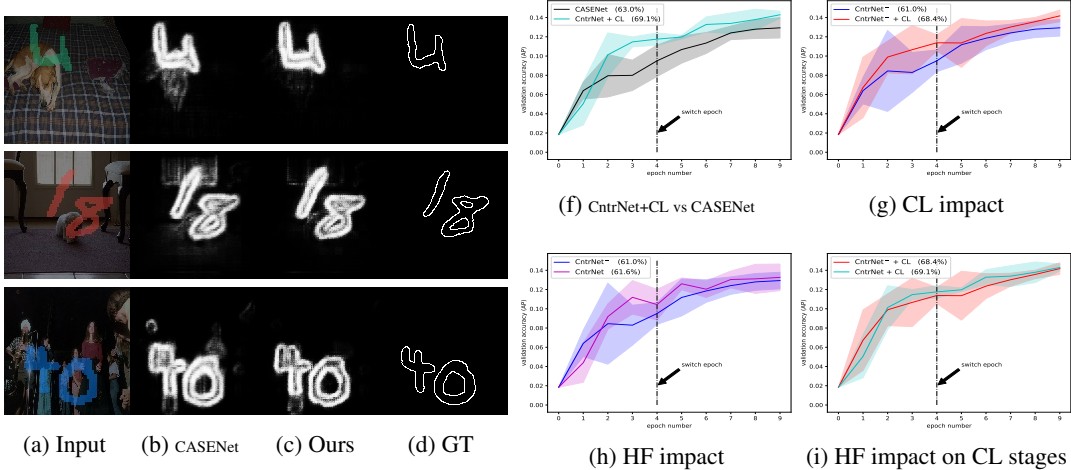

(a) Input    (b) CASENet    (c) Ours    (d) GT

(f) CntrNet+CL vs CASENet    (g) CL impact

(h) HF impact    (i) HF impact on CL stages

Figure 4: Evaluation results on contourMNIST

**Implementation details**: We have implemented all our methods using the PyTorch machine learning framework. We used SGD optimizer with learning rate $1e - 4$ for all experiments. Training experiments are done using NVIDIA GTX-1080 GPU with 12GB RAM. For wavelets decomposition, we used pytorch-wavelets[1] package. All the models in this section are trained for ten epochs. Switch epoch is set to 5th epoch for CL-based training experiments. We refer our high-frequency augmented ConvNet as *CntrNet* for brevity in this section.

**Evaluation criteria**: We used average precision (AP) as an evaluation metric. Prediction maps are thresholded with a fixed threshold of $0.5$ to compute false and true positives. The usual groundtruth labels ($Y$) are used for validation. For simplicity, we omitted expensive post-processing steps (such as thinning) and evaluations at optimal-data-scale [ODS Acuna et al. (2019)] during training epochs. Each training experiment is run for five times for statistical significance of the results and observations. Validation accuracy, in terms of AP, is computed for every epoch and plotted in Figure 4. Shaded regions in the plot represent standard deviations in the validation accuracies of different models and training schemes. For final trained models, we also report AP measures at ODS, i.e. maxAP, that finds maximum AP obtained by different thresholds. These values are shown in the brackets, following the label names in the legends of the plots.

---

[1] www.pytorch-wavelets.readthedocs.io

### 4.1.2 ABLATION STUDIES

**How do our proposals (*CntrNet+CL*) perform compared to the state-of-the-art?**
We first start comparing the performance of our proposals with a state-of-the-art method, CASENet [Yu et al. (2017)]. Figure 4e illustrates the validation accuracy plots of CASENet and our *CntrNet+CL* over training epochs. *CntrNet+CL* seem to converge faster than the CASENet. According to maxAP scores mentioned in the legend of the plot, our network (*CntrNet+CL*) outperforms CASENet by approximately 6%. It means that *CntrNet+CL* produces less false alarms than the CASENet. This is also quite evident in the qualitative results provided in Figure 4b-c.

**Is the proposed CL training necessary?**
Here we evaluate how curriculum learning impacts the learning processes in the absence of high-frequency (HF) augmentation. Towards this, we experiment with CntrNet without HF augmentation,*CntrNet⁻*. Validation accuracy plots are as shown in Figure 4f when *CntrNet⁻* is trained with and without CL-based scheme. The model achieves better generalization performances when trained with CL. It improves validation accuracy by 7.4 % compared to the one trained without CL.

**Is the proposed high-frequency augmentation necessary?**
We also evaluate how high-frequency augmentation impacts the model learning process and performance in the absence of curriculum learning. As seen in Figure 4g, high-frequency augmentation seems to help the model to achieve better performances promptly, starting from Stage 1. However, it seems to be helping the model's performance by only a small extent (0.6%) in the end.

**How does high-frequency augmentation impact the first stage of CL?**
As seen in Figure 4i, the explicit augmentation of high-frequencies seems to be helping stage 1 to converge faster. This is expected because Stage 1 is about identifying the pixels with high frequencies and recognize their class.

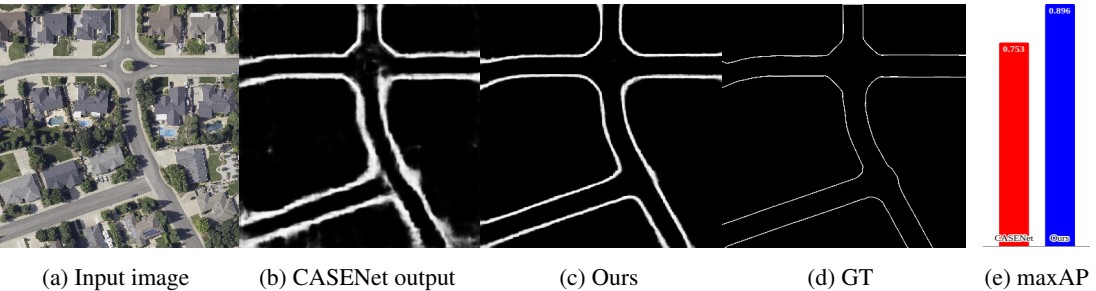

(a) Input image      (b) CASENet output      (c) Ours      (d) GT      (e) maxAP

Figure 5: Evaluation results on Areal Image Road Contours dataset

### 4.2 RESULTS ON AERIAL ROAD CONTOURS DATASET

In this section, we evaluate the models on a real-world aerial image dataset for the task of extracting road contours.

**Dataset preparation**: Our dataset originally is prepared for geo-parcel alignment task (Section 5) and contains 13,368 aerial image tiles captured over city of Redding, California. Each of these tiles is $4864 \times 7168$ resolution and approximately cover an area of $0.15 \times 0.2$ square miles. Samples of these tiles are shown in Figures 1 and 7. For training and evaluation experiments, we manually label 11 of these aerial image tiles with road contours. This labeling process took approximately 2 hours per tile. Ten tiles are used to prepare a training set, and one tile is left for validation. To prepare our train and val sets, we randomly crop several subregions of $2000 \times 2000$ from these tiles and resize them to $256 \times 256$. We also use some data augmentation techniques to amplify the scale of the set. The train and val sets contain 21649 and 522 samples, respectively.

**Implementation**: The network's implementation details are similar to above. Here we set batch size as eight and input size is $256 \times 256$. The models are trained for 20 epochs that take approximately 2 days 18 hours on AWS GPU instance with NVIDIA Tesla K80.

**Evaluation**: We report maxAP scores for both CASENet and our model (CntrNet+CL) in Figure 5e. Our method performs with an accuracy of 89.6%, while CASENet is at 75.3% maxAP points. In

other words, our method outperforms CASENet by nearly 15%. As shown in Figure 5, our results are sharper and accurate compared to CASENet. A result at the tile scale can be seen in Figure 6. We use this model in a real-world application of geo-parcel alignment in the next section.

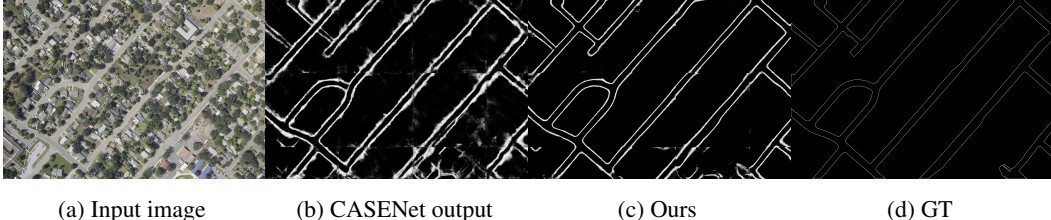

(a) Input image         (b) CASENet output         (c) Ours         (d) GT

Figure 6: Sample result on a tile from Parcel alignment dataset

## 5 APPLICATION TO GEO-PARCEL ALIGNMENT

In this section, we discuss an application of the proposed model, *CntrNet+CL* (trained for road contour extraction), for aligning *geo-parcel* data with aerial image tiles. Geo-parcel data is generally used to identify public and private land property boundaries for tax assessment processes. Parcels are shapefiles in the records maintained by local counties and represent *latitude-longitude* GPS coordinates of the property boundaries. We project these parcel shapes (using perspective projection) onto a geo-tagged coordinate system of the camera with which aerial imagery was captured. This process results in binary contour images as shown in Figure 7c. These contours are ideally expected to match visual property entities in the aerial image of the corresponding region (shown in Figure 7a). However, due to several differences in their collection processes, these two modalities of the data often misalign by a large extent, sometimes, in the order of 10 meters. Figure 7d depicts the misalignment of the original (before alignment) parcel contours overlaid on the aerial image in blue color. This misalignment might lead to wrong property assignments to individuals, thus, incorrect tax assessments. These data modalities of geographical data need to be aligned well before using it to assist the processes of property assignment and tax assessment.

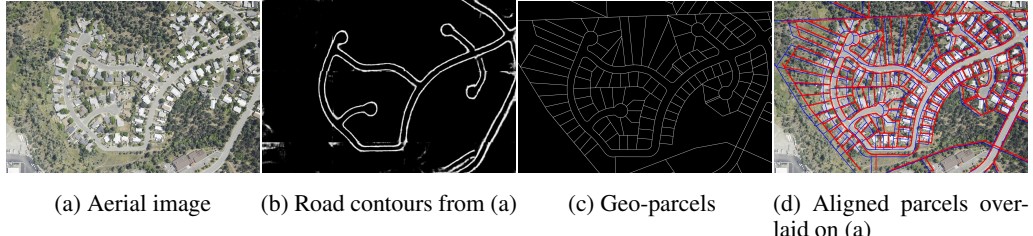

(a) Aerial image         (b) Road contours from (a)         (c) Geo-parcels         (d) Aligned parcels overlaid on (a)

Figure 7: Geo-parcel alignment with road evidences. Better see in enlarged version.

We use our *CntrNet+CL* model trained on the aerial road contours dataset prepared in Section 4.2. Given an aerial image tile, as shown in Figure 7a, we first divide the tile into 12 non-overlapping patches and pass them as a batch to the model. The predictions from the model are stitched back to form a road contour prediction at tile resolution, as shown in Figure 7b. One can pose this alignment problem as an image registration task by considering geo-parcel and road contour images as moving and target images, respectively. We designed an image registration network for the same, and discussions about the registration network is out of the scope of this paper. A few samples of final aligned parcels are overlaid with red color in Figure 7d.

## 6 CONCLUSIONS

In this work, we presented a novel ConvNet with explicit high-frequency augmentation and a new two-stage curriculum learning scheme for class-specific object boundary extraction. Our ablation studies with simulated MNIST digit-contours dataset demonstrated that this explicit high-frequency augmentation helps the model to converge faster. Our high-frequency augmented model, when trained with proposed CL based scheme, outperformed CASENet by nearly 15% on aerial image dataset. We also demonstrated a use-case of developed contour extraction model to align geo-parcel boundaries with roads extracted from aerial imagery.

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
