# OpenReview forum: "High-Frequency guided Curriculum Learning for Class-specific Object Boundary Detection"
_ICLR.cc/2020/Conference — Reject_

### Official Review · AnonReviewer2 · 2019-10-23
**Official Blind Review #2**

**Rating:** 1

**Review:**

Summary: The suggest two improvements to boundary detection models: (1) a curriculum learning approach, and (2) augmenting CNNs with features derived from a wavelet transform. For (1), they train half of the epochs with a target boundary that is the intersection between a Canny edge filter and the dilated groundtruth. The second half of epochs is with the normal groundtruth. For (2), they compute multiscale wavelet transforms, and combine it with each scale of CNN features. They find on a toy MNIST example that the wavelet transform doesn’t impact results very much and curriculum learning seems to provide some gains. On the Aerial Road Contours dataset, they find an improvement of ~15% mAP over the prior baseline (CASENet).

I have several concerns with this work:
* The idea of using wavelet transforms to augment CNNs has been more thoroughly explored in prior work (e.g., see [1]).
* No comparison to existing SOTA segmentation models (e.g., [2]). These semantic / instance segmentation models can easily be adapted to the task of boundary detection. I suspect the baseline here is weak.
* Section 6 is severely unfinished. The explanation is sparse and there are no quantitative results -- just the output of the model overlaid on one example.
* The choice of curriculum learning task is arbitrary, and there are no ablations explaining why this is a reasonable task. For example, what about random subsets of pixels? At the moment, it offers no insight for practitioners.
* There are no ablations for the Aerial Road Contours experiments. This seems necessary because it is the only realistic dataset evaluated in this work. The MNIST experimental results appear qualitatively different from the Contours experiment. For example, they show that wavelet features do not make much of a difference, but does it make a difference for Contours?

Altogether, this work unfortunately offers few insights to vision practitioners, let alone general practitioners. Substantial work needs to be devoted to expanding experimental coverage.

[1] Wavelet Convolutional Neural Networks. Shin Fujieda, Kohei Takayama, Toshiya Hachisuka
[2] TensorMask: A Foundation for Dense Object Segmentation. Xinlei Chen, Ross Girshick, Kaiming He, Piotr Dollár

**Experience Assessment:**

I have published one or two papers in this area.

**Review Assessment: Checking Correctness Of Derivations And Theory:**

N/A

**Review Assessment: Checking Correctness Of Experiments:**

I carefully checked the experiments.

**Review Assessment: Thoroughness In Paper Reading:**

I read the paper thoroughly.

---

### Official Review · AnonReviewer1 · 2019-10-28
**Official Blind Review #1**

**Rating:** 1

**Review:**

The main idea of the paper is adding a curriculum learning-based extension to CASEnet, a boundary detection method from 2017. In the first phase, the loss emphasizes easier examples with high gradient in the image, and in the second phase, the method is trained on all boundary pixels. This change seems to improve edge detection performance on a toy MNIST and an aerial dataset.

A second innovation claimed by the authors is adding Wavelet decomposition-based processing into the net. Unfortunately, this mostly only speeds up learning, as the ablation does not show meaningful improvements relative to the error bounds in later stages of training. Furthermore, the paper lacks a discussion of related work on incorporating wavelet ideas into neural networks. For example:
-- Generic Deep Networks with Wavelet Scattering, by Ouyallon et al.
-- Invariant scattering convolution networks, by Bruna et al
and multiple more recent ones. Without either clear performance gains or more in-depth discussion of this novelty, it is not clear how to take it into account.

When reading the paper, it appears that "boundary detection" for the cases that the authors are exploring is very directly related to 2-class semantic segmentation (road / non-road), the only difference being that the edge boundaries are weighted much higher in the cross-entropy loss. As such, there is a lot more recent net architecture work for semantic segmentation that should be directly applicable, and should perform much better than CaseNet when adapted to the task. As a result, the experiments and the significance of this paper are rather marginal.

In experimental results, the authors threshold prediction with 0.5, which is suboptimal. The resulting metric, which is just "accuracy" is called incorrectly "average precision". Instead, true definition of average precision should be used, that is not dependent on potentially suboptimal fixed thresholding but on the area under precision-recall curve instead. Finally, it would be helpful to do ablation and confidence bounds also on the main aerial road results, as the 15% gain is significantly more than the gain that appears in the toy dataset.

**Experience Assessment:**

I have published in this field for several years.

**Review Assessment: Checking Correctness Of Derivations And Theory:**

I carefully checked the derivations and theory.

**Review Assessment: Checking Correctness Of Experiments:**

I carefully checked the experiments.

**Review Assessment: Thoroughness In Paper Reading:**

I read the paper thoroughly.

---

### Official Review · AnonReviewer3 · 2019-10-29
**Official Blind Review #3**

**Rating:** 3

**Review:**

The paper shows the efficiency of curriculum learning and using problem specific features for contour detection.

The authors consider a network trained for class-specific edge detection (e.g. outlining edges of roads in an image). They propose two problem domain tricks to improve the performance:
- use curriculum learning by training the network to first detect the "easy" edges, i.e. edges found also using the Canny edge detector
- add high frequency wavelet coefficients as additional feature maps to the convnet.

The two techniques prove important on two tasks:
- modified MNIST edge detection
- road boundary detection in aerial imaginery.

Maybe the most important aspect of the paper is that shows that with little data (the real world aerial imaginary dataset had only 11 labeled tiles) manual feature engineering and smart cost function selection are still relevant.
Since this is a common pattern in many application domains, such as specialized medical image processing where labeled data is scarce, the paper is important. However, it is not clear if the proposed chanes are needed when more labeled data is available and how much do they overfit to the small test set.

The paper could be strengthened by analysing the impact of the proposed problem-dependent CL and features versus the amount of available training data. Are they still relevant with 100 labeled images?
These experiments could be even run on the artificial MNIST set.

Moreover, some analysis of result significance is needed. On the real-world dataset there is only 1 test case!! How much was the network tuned to properly work on it? Maybe the authors can run a  cross-validation to show that the results don't overfit to this one test image?

Minor remarks: You refer to Stephane's Mallat book as Stephane 1999, this is wrong, his first name is Stephane and last is Mallat, please fix the bibliography and use Mallat 1999.


**Experience Assessment:**

I have read many papers in this area.

**Review Assessment: Checking Correctness Of Derivations And Theory:**

I assessed the sensibility of the derivations and theory.

**Review Assessment: Checking Correctness Of Experiments:**

I assessed the sensibility of the experiments.

**Review Assessment: Thoroughness In Paper Reading:**

I read the paper at least twice and used my best judgement in assessing the paper.

---

### Decision · Program_Chairs · 2019-12-19

**Decision:**

Reject

**Comment:**

This paper received all negative reviewers, and the scores were kept after the rebuttal. The authors are encouraged to submit their work to a computer vision conference where this kind of work may be more appreciated. Furthermore, including stronger baselines such as Acuna et al is recommended.